# Stimulation of *Arabidopsis thaliana* Seed Germination at Suboptimal Temperatures through Biopriming with Biofilm-Forming PGPR *Pseudomonas putida* KT2440

**DOI:** 10.3390/plants13192681

**Published:** 2024-09-24

**Authors:** Chandana Pandey, Anna Christensen, Martin N. P. B. Jensen, Emilie Rose Rechnagel, Katja Gram, Thomas Roitsch

**Affiliations:** 1Department of Plant and Environmental Sciences, Copenhagen Plant Science Centre, University of Copenhagen, 1172 Copenhagen, Denmark; ac@plen.ku.dk (A.C.); martinpalaciosg@gmail.com (M.N.P.B.J.); kfn800@alumni.ku.dk (E.R.R.); dbw978@alumni.ku.dk (K.G.); roitsch@plen.ku.dk (T.R.); 2Global Change Research Institute of the Czech Academy of Sciences, 60300 Brno, Czech Republic

**Keywords:** biostimulant, beneficial microbe, biofilm, arabidopsis, ecotypes, temperature, germination, rhizobacteria, biopriming

## Abstract

This study investigated the germination response to temperature of seeds of nine *Arabidopsis thaliana* ecotypes. They are characterized by a similar temperature dependency of seed germination, and 10 °C and 29 °C were found to be suboptimal low and high temperatures for all nine ecotypes, even though they originated from regions with diverse climates. We tested the potential of four PGPR strains from the genera *Pseudomonas* and *Bacillus* to stimulate seed germination in the two ecotypes under these suboptimal conditions. Biopriming of seeds with only the biofilm-forming strain *Pseudomonas putida* KT2440 significantly increased the germination of Cape Verde Islands (Cvi-0) seeds at 10 °C. However, biopriming did not significantly improve the germination of seeds of the widely utilized ecotype Columbia 0 (Col-0) at any of the two tested temperatures. To functionally investigate the role of KT2440’s biofilm formation in the stimulation of seed germination, we used mutants with compromised biofilm-forming abilities. These bacterial mutants had a reduced ability to stimulate the germination of Cvi-0 seeds compared to wild-type KT2440, highlighting the importance of biofilm formation in promoting germination. These findings highlight the potential of PGPR-based biopriming for enhancing seed germination at low temperatures.

## 1. Introduction

Seed germination is a crucial stage in plant development, signifying the transformation of quiescent seeds into actively growing seedlings and initiating the plant life cycle [1,2]. The quality, speed, and uniformity of germination influence plant growth and overall health [3,4]. It is, therefore, important to understand the factors influencing seed germination in order to optimize plant productivity, especially in challenging agricultural and ecological contexts [5].

Seed biopriming techniques, involving the application of microorganisms and signaling molecules like hormones through seed treatments, have gained increasing attention for improving plant establishment under suboptimal environmental conditions [6]. These biopriming techniques facilitate the interaction between the seeds’ surface and microorganisms, leading to accelerated hydration and modulation of microbe-mediated signaling molecules, which ultimately enhance plant growth. The intricate matrix of seed biology involves a dynamic interplay between the natural microbial communities residing on the seed exterior. The microorganisms infiltrate the seed, resulting in shorter imbibition times and, consequently, higher germination rates [2,7,8]. Plant Growth-Promoting Rhizobacteria (PGPR) are valuable biostimulants known for their potential to avoid or reduce yield losses by improving resilience against stress conditions. PGPR applications have been reported to increase crop yields, typically ranging from 10% to 20% [7]. Biopriming seeds with a diverse range of PGPR have emerged as a potent strategy to enhance plant resilience when plants face challenging environmental conditions [9,10], thus providing protection on demand. 

The soil, being a complex ecosystem, plays a critical role in the process of seed germination. Microbial symbionts and abiotic stress factors substantially influence the soil microbial communities and crop production. Soil bacteria in the rhizosphere contribute to nutrient provision, making microbial fertilizers with PGPR increasingly attractive for sustainable agriculture [11,12]. The application of PGPR in agricultural practices aims to alleviate adverse environmental conditions and enhance crop productivity [13]. Several studies have highlighted PGPR’s positive impact on plant growth and yield through direct and indirect mechanisms [14,15,16,17,18]. These mechanisms include the release of plant growth regulators (phytohormones), ability to form biofilms, and improvement of nutrient accessibility [14,19,20]. Certain bacterial strains have the ability to convert both insoluble inorganic and organic nutrients into soluble forms, thereby making them accessible to enhance seedling development and overall plant health. This conversion is achieved through processes like solubilization and mineralization [21,22,23]. PGPR strains like *Pseudomonas putida* KT2440, known for biofilm formation, benefit various crops by establishing bacterial communities on plant surfaces [18,24]. 

Biopriming with PGPR has been shown to enhance plant resilience by promoting shoot growth, facilitating nutrient uptake, synthesizing growth-promoting substances, and fortifying plants against stressors. PGPR-mediated biopriming offers a promising strategy for overcoming challenges during seed germination and boosting crop yields [14,25]. Arabidopsis ecotypes show adaptive differentiation in response to environmental cues [26]. Understanding how developmental responses vary with temperature fluctuations in the environment is crucial for elucidating the impact of PGPR on the initial phase of plant development. This study aims to determine the temperature dependency of seed germination in nine *A. thaliana* ecotypes adapted to a range of environments. The chosen ecotypes originate from contrasting ‘home’ environments, ranging from high alpine regions to the early spring season and hot, dry climate. Notably, our study investigated the impact of PGPR on seed germination under suboptimal low- and high-temperature conditions, with the aim of advancing sustainable seed priming techniques. The selected PGPR strains for the investigation included *P. putida* KT2440, known for its ability to produce biofilm and enhance the growth and plant defenses in tomato and maize [27,28,29]; *P. fluorescens* SS101, recognized for promoting growth and disease resistance in tomato against *Phytophthora infestans* [30]; *P. putida* UW4, which produces the plant hormone auxin and supports canola growth and *B. amyloliquefaciens* FZB [31,32,33], renowned for increasing both height and fresh weight in tobacco and lettuce. Lastly, our research aims to explore the influence of the potential for the formation of biofilms on seed germination, notably utilizing available mutant strains of KT2440 bacteria with reduced biofilm formation. Our goal was to functionally address the relationship between biofilm formation and the ability to stimulate seed germination. 

## 2. Results

### 2.1. Temperature Dependency of Seed Germination of A. thaliana Ecotypes from Diverse Growth Habitats

In this study, we aimed to understand how distinct Arabidopsis ecotypes respond to varying temperatures during seed germination. We used nine ecotypes, Cvi-0 (Cape Verde Islands), Col-0 (Columbia), Bur (Burren), C24 (Coimbra), Bla-1 (Blanes), Neo-6 (Shahdara), N13 (Konchezero), Dja-1 (Djarly), and MS0 (Moscow), adapted to different natural habitats. Our investigation explored the influence of temperature on germination, elucidating the optimal and suboptimal low or high temperatures for germination across all nine *A. thaliana* ecotypes (Table 1).

The experimental approach is described in Section 4 and illustrated in Figure 1.

Considering the different native habitat temperature conditions of the nine ecotypes tested, we have investigated seed germination at four different temperatures 10, 17, 24, and 29 °C, as shown in Figure 2. Our results showed that seed germination was generally reduced at 10 °C, thus identified as a suboptimal low temperature, while increased germination was observed in all nine investigated ecotypes at temperatures of 17 °C and 24 °C, representing optimal temperatures. Further increasing the temperature to 29 °C significantly reduced the seed germination of all ecotypes, thus representing a suboptimal high temperature compared to those at 17 °C and 24 °C. Therefore, polynomial curves for the temperature dependency of germination had the lowest points in the curve at 10 °C and 29 °C. Interestingly, at 10 °C, five ecotypes (C24, Bla-1, Neo-6, Dja-1, and MS0) out of nine had higher germination percentages than those at 29 °C, as shown in Figure 2. The germination of Cvi-0, C24, and MS0 was significantly greater at 17 °C than at 24 °C, while the reverse was true for Bur. Germination did not differ between the two temperatures for the other five ecotypes (Figure 2). This finding highlights the sensitivity of these genotypes to temperature variations, with suboptimal temperatures negatively impacting their germination. The optimum temperature for seed germination for each ecotype was calculated by analyzing the peak point of the polynomial curve. Bur and N13 had a higher optimal temperature than Bla1, which had the lowest optimal temperature (Appendix A). Specifically, Bur had an optimal temperature of 20.6 °C, while Bla-1 had the lowest optimal temperature at 17.6 °C.

Regardless of their natural habitat, ranging from cold to warm climates, all the selected ecotypes had suboptimal temperatures of 10 and 29 °C. This highlights that both 10 °C and 29 °C can generally be considered suboptimal low and high temperatures in the context of Arabidopsis seed germination. Furthermore, among the nine ecotypes, Arabidopsis Col-0, which is widely utilized in research, and Cvi-0, which is adapted to hot, dry climates, were specifically chosen for further investigation to assess the impact of PGPR. This selection was based on similar low germination rates at suboptimal temperatures of 10 °C and 29 °C, despite originating from diverse environmental backgrounds.

### 2.2. At Suboptimal Low Temperatures, the PGPR Strain KT2440 Stimulate Seed Germination of the A. thaliana Genotype Cvi-0

The assessment of the temperature dependency of the nine selected ecotypes revealed that 10 °C and 29 °C are suboptimal low and high temperatures, respectively. Thus, we selected two ecotypes: Col-0, adapted to the moderate climate of Columbia (USA), and Cvi-0, adapted to the hot and dry climate of the Cap Verde Islands. Considering the results of the temperature dependency of seed germination, our subsequent experiments were designed to explore the impact of different bacteria on seed germination at a suboptimal low temperature of 10 °C and a suboptimal high temperature of 29 °C at 7 days after inoculation (dai), as shown in Figure 3. We investigated the impact of four different bacteria, *Pseudomonas fluorescens* SS101 (SS101), *Pseudomonas* sp. UW4 (UW4), *Bacillus amyloliquefaciens* FZB (FZB), and *Pseudomonas putida* KT2440 (KT2440) are known for their plant growth-promoting characteristics.

Results showed a significant increase in seed germination of Cvi-0 in the presence of KT2440 at 10 °C compared to the mock control treatment. However, the positive impact of KT2440 was only evident in Cvi-0, as it did not lead to a higher germination in Col-0. Inoculation with the other three bacterial strains did not result in any significant changes in the germination of Col-0 and Cvi-0 at 10 °C. Despite their growth-promoting properties, the application of PGPRs did not induce significant changes in germination at a suboptimal high temperature of 29 °C. These results highlight the substantial differential impact of PGPR application on the germination of ecotypes originating from different habitats. This highlights the specificity of the PGPR response in different genotypes of the same species and also indicates the importance of plant-derived factors within the interaction with the microbes, which probably reflect the plant’s adaptation to different habitats. 

### 2.3. Effect of PGPR Strain KT2440 and Its Mutants on Cvi-0 Seed Germination under Suboptimal Low Temperature

In our screening, we observed a significant increase in the seed germination of Cvi-0 at suboptimal low temperatures when exposed to the biofilm-forming bacterium KT2440. Thus, this interesting finding on the impact of the dynamics of seed germination at lower temperatures could be due to the formation of a biofilm by KT2440. Considering this observation, we performed an experiment to functionally address the mechanism governing the ability of KT2440 to influence seed germination. This study aimed to elucidate the correlation between biofilm formation and seed germination using biofilm-deficient mutant bacteria, specifically KT *alg*^−^*bcs*^−^*pea*^−^*peb*^−^ (KT q) and KT *alg*^−^ [23]. KT2440, in its quadruple mutant form KT q, has all four potentially identifiable exopolysaccharide gene clusters (alg^+^, bcs^+^, pea^+^, and peb^+^) deleted, while the KT *alg*^−^ mutant specifically lacks the ability to synthesize exopolysaccharide alginate [26].

We investigated whether bacteria with compromised biofilm-forming capabilities (KT q and KT *alg*^−^) differed from the wild-type strain KT2440 in terms of their impact on seed germination at a suboptimal temperature of 10 °C. Seed germination from the mock treatment was compared to that from treatments with KT2440 and its mutants KT q and KT *alg*^−^, as shown in Figure 4.

It was hypothesized that any alterations in biofilm formation would negatively influence the plant-PGPR interaction, resulting in reduced germination rates. Our results showed that KT2440 significantly stimulated the seed germination percentage at 7 dai and 10 dai compared to the mock treatment (Figure 4). In contrast, KT q showed a significant decrease in the germination percentage at the respective time points. KT *alg*^−^, however, did not demonstrate any significant changes compared to the mock treatment. 

## 3. Discussion

The natural variability observed within ecotypes of the same species, acclimated to varying pedoclimatic conditions, provides a valuable framework for examining how PGPR impacts the dynamic interplay among genetic factors and environmental influences, especially during the early stages of the life cycle. In this study, our focus was on investigating the temperature dependency of *A. thaliana* seed germination using a set of nine ecotypes adapted to different climates. The results we obtained helped us gain a deeper understanding of how KT2440 stimulates seed germination compared to mutants, addressing questions about the specific mechanism used by PGPRs to stimulate seed germination under suboptimal low-temperature conditions and facilitate to develop solutions for sustainable plant growth in environments characterized by temperature limitations. 

### 3.1. Impact of Temperature on Seed Germination 

This study focused on the temperature dependency of seed germination across diverse *A. thaliana* ecotypes to gain insights into the dynamics of germination influenced by temperature. By examining nine ecotypes from different natural habitats, our approach aimed to unravel the germination response of *A. thaliana* to temperature variation and lay a foundation for understanding their germination under diverse environmental conditions.

According to the findings of Huang et al. [34], different *A. thaliana* ecotypes have evolved distinct traits influenced by their respective environmental conditions. Our investigation revealed the highest germination rates among all ecotypes at temperatures between 17 °C and 24 °C, regardless of their diverse habitats. This suggests a consistent germination response to temperature among the different ecotypes, indicating similar temperature dependence of germination. Interestingly, the widely used Col-0 and other ecotypes from extreme environments had similar optimum temperatures, indicating a predisposition for maximum germination at an ambient range of temperatures. Cvi-0, adapted to a hot and dry climate, and Bur, adapted to a cool and damp climate, exhibited winter and summer annual phenotypes, respectively [34]. Despite their distinct climates, both Cvi-0 and Bur shared the same optimum temperature for germination. This observation highlights the potential role of specific temperature conditions in driving evolutionary differences [35,36], potentially leading to the emergence of new ecotypes within a particular plant species [37]. However, at 29 °C, seed germination decreased. This aligns with the previous findings, which were conducted to compare the germination of two ecotypes, Cvi-0 and Bur, at 10 °C and 25 °C [24]. Optimum temperatures of 17 °C and 24 °C had higher germination rates compared to suboptimal temperatures of 10 °C and 29 °C, respectively.

In summary, our findings suggest that seed germination can occur at lower suboptimal temperatures, possibly reflecting the transitional period from winter to spring in temperate regions. The highest germination rates were observed at 17 °C to 24 °C, thus representing the optimum temperature across ecotypes. Delayed germination at higher or lower temperatures may result in diminished yield and decreased competitive edge, which can potentially be mitigated by faster germination at suboptimal temperatures, particularly through the application of PGPR [38,39].

### 3.2. Impact of PGPR on Seed Germination Responses at Suboptimal Low and High Temperature

Bioinoculants have emerged as a highly effective strategy for promoting sustainable agriculture, with the potential to exert a substantial impact on both global food security and the self-sufficiency of individual nations [18,40,41]. 

These bioinoculants are carefully developed from a diverse array of PGPR strains, each possessing a remarkable capacity to stimulate plant growth and, in turn, strengthen crop yields, even under adverse environmental conditions [2]. These PGPRs show the ability to synthesize a wide spectrum of essential components, encompassing plant growth regulators, enzymes, inducers, extracellular polysaccharides, metallophores, antibiotics, volatile compounds, peptides, bacteriocins, and numerous other secondary metabolites [42,43], enabling them to adapt to challenging environmental contexts. Despite extensive research on the impact of PGPR on crop growth, there has been limited focus on early time points and intricate interactions between plant seeds and PGPRs. All four selected strains demonstrated the following attributes: production of plant growth regulators, enzymes, extracellular polysaccharides, and antibiotics. KT2440, a gram-negative strain, is also known to produce a dominant biofilm, metallophores, and various volatile compounds, while FZB is known for its production of diverse antibiotics and VOCs. Therefore, initiating investigations with germination tests is crucial, as well as using straightforward and efficient methods to evaluate multiple bacterial strains, especially for large-scale studies. Subsequently, bacteria that show the most promising results should undergo comprehensive characterization to unveil the mechanisms underlying their roles in promoting plant growth. Traditional seed germination methods are conventionally used to assess the seed lots’ physiological quality in various aspects of plant and seed production. However, there is still a substantial gap in standardized techniques for evaluating the performance of PGPR and their interactions with germination responses under specific extreme conditions while maintaining sterility. Within our study, we have established a standardized germination assay as an experimental basis for screening bacteria for their potential as priming agents to enhance germination in the seeds of a model plant. This study established a simple and robust method to test seed germination and evaluate the impact of microbial strains. We found that the application of KT2440 resulted in an improvement in Cvi-0 seed germination under suboptimal low-temperature conditions.

Previous studies have shown that combining cold stratification with biopriming using PGPR strains from *Pseudomonas* and *Bacillus* promotes hazelnut seed germination in other plant species [44]. Similarly, another investigation revealed that using hydropriming and biopriming with *Pseudomonas* and *Bacillus* significantly improves germination of *Abies hickelii* and *Abies religiosa* in soil [45]. The application of KT2440 resulted in an increase in seed germination at 7 dai and 10 dai under suboptimal low-temperature conditions. It is important to highlight that significant improvements were observed when comparing KT2440 with other *Pseudomonas* and *Bacillus* strains. SS101, UW4, and FZB had either no significant effect or less influence compared to both the KT2440 strain and the mock treatment control. This difference may be because KT2440 can form a biofilm, which may affect other plant hormone functions and promote seed germination [42,46,47]. Although *Pseudomonas* and *Bacillus* are known to promote plant growth through the production of growth regulators, enzymes, and antibiotics, these properties were not assessed in this study. However, KT2440 had a significant ability to affect germination under suboptimal low temperatures, broadening our understanding of its impact under more extreme and less-studied conditions.

### 3.3. The Significance of KT2440’s Biofilm-Forming Ability for Seed Germination

In this study, KT2440 significantly improved the germination of Cvi-0 at 7 dai and 10 dai compared to the control, KT q, and KT *alg*^−^ treatments under suboptimal low-temperature conditions. To investigate the role of the biofilm in this process, we included two KT2440 mutants with impaired biofilm formation: KT q and KT *alg*^−^. Previous research has highlighted the importance of biofilms in enhancing biotic stress resistance in *A. thaliana* [46,48]. KT q is a quadruple mutant in which all four potentially identifiable exopolysaccharide gene clusters (alg^+^, bcs^+^, pea^+^, and peb^+^) have been deleted. These gene clusters are responsible for producing specific exopolysaccharides. KT *alg*^−^ has been genetically altered to lack the ability to synthesize a particular exopolysaccharide, known as alginate. Alginate is a type of linear polymer that is commonly produced by certain bacteria and has various functions, including contributing to biofilm formation.

Our study supports that biofilm formation is crucial for the beneficial impact of KT2440 on Cvi-0 germination under suboptimal low temperatures. Specifically, the KT q mutant inoculation with Cvi-0 resulted in a significant decrease in germination compared to both KT2440 and the mock treatment at 10 °C at 7 dai and 10 dai (Figure 4). Interestingly, KT *alg*^−^ appeared to have a higher potential to stimulate seed germination percentage than the KT q mutant, which is consistent with previous findings that KT q compromises biofilm stability [26]. Other *P. putida* strains known to promote plant growth in challenging environments include GAP-P45 strains, which have been shown to promote growth and mitigate drought stress in maize and sunflower plants [49], potentially through the production of exopolysaccharides. The ability of UW4 to promote canola growth under saline conditions by reducing salt-induced ethylene synthesis is actually attributed to its ACC deaminase activity [50]. Although the exact mechanisms underlying these stimulating effects have not been fully elucidated, they may involve the formation of protective root barriers, competitive advantages in the rhizosphere, and quorum sensing for rapid adaptation to changing conditions. The observation that the ability of KT2440 to stimulate seed germination in Cvi-0 under suboptimal conditions is compromised in a biofilm mutant suggests an important role for biofilm in positively influencing the ability to stimulate seed germination. The approach with mutants demonstrates that the biofilm, which facilitates interactions between the bacteria and seeds, is functionally relevant in promoting optimal conditions for germination, which may be involved in improved seed hydration and stress protection. This insight highlights the importance of biofilm formation as a beneficial factor in the complex dynamics of plant-bacterial interactions. Biofilm-producing PGPR strains, such as KT2440, hold potential for future agricultural use due to their valuable traits, including effective root colonization, improved germination, and resilience against environmental stresses.

KT2440 was only able to stimulate seed germination in ecotype Cvi-0 but not in Col-0. The genotypic differences in the germination response of the two ecotypes could have various reasons and be potentially related to intrinsic differences between the two ecotypes, related to their distinctly different habitats characterized by specific pedoclimatic conditions. These differences could involve different traits, such as variations in seed coat permeability, metabolic profiles, hormonal balances, or endophytes. This finding is of practical relevance to the development of robust biostimulants for crop plants. A major problem with biostimulants is the lack of robustness across cultivars of a particular crop species. Thus, the observed differences in plant responsiveness could be used as a basis to identify genetic determinants, which can then be established as novel breeding targets for more robust performance of beneficial microbes applied as biostimulants.

## 4. Materials and Methods

### 4.1. Seed Material and Growth Conditions

In this study, we used nine ecotypes, Cvi-0, Col-0, Bur, C24, Bla-1, Neo-6, N13, Dja-1, and MS0. These ecotypes originate from regions with diverse climates, including high alpine environments characterized by cooler temperatures and shorter growing seasons, as well as early spring climates with fluctuating temperatures, as shown in Table 1. For example, Cvi-0, from the Cape Verde Islands, is adapted to a tropical climate with mild temperatures, while Col-0, one of the most widely studied ecotypes, comes from Columbia, Missouri, representing a temperate climate. Other ecotypes, such as Bur, C24, Bla-1, Neo-6, N13, Dja-1, and MS0, originated from various parts of Europe and Asia, where they have adapted to different altitudes and latitudes, ranging from cooler, high-altitude regions to warmer, low-altitude areas (Table 1). These ecotypic differences reflect the plants’ genetic adaptations to their specific environments, influencing their responses to factors like temperature and moisture availability. Understanding these variations is crucial for studying how plants adapt to changing climates and for developing strategies to enhance crop resilience. Approximately 10 mg of seeds from each ecotype was used for a series of sterilization steps. Initially, 400 µL of Milli-Q water (MQ water) was added to each tube containing seeds for two min., followed by the removal of water and subsequent treatment with 200 µL of 70% ethanol for two min. The ethanol was then removed and the seeds were washed three times with 400 µL of MQ water. Subsequently, the seeds were sterilized once more, this time with 200 µL of 10% NaOCl (Sodium hypochlorite) for two min. Following this, the seeds were washed three times with 400 µL MQ water. After these procedures, 1 mL of autoclaved MQ water was added, and the tubes were sealed with parafilm before being stored at 4 °C for a period of two days.

All the ecotypes were subjected to germination at four distinct temperatures (10 °C, 17 °C, 24 °C, and 29 °C) to assess their responses under suboptimal temperature conditions. Furthermore, among the nine ecotypes, Arabidopsis Col-0, which is widely utilized in research, and Cvi-0, adapted to hot, dry climates, were specifically chosen for further investigation with PGPR at both 10 °C and 29 °C. The seed germination (the protrusion of the radicle) was assessed at 7 dai.

### 4.2. Seed Germination Assay at Four Distinct Temperatures

Surface-sterilized seeds were placed in autoclaved Eppendorf tubes containing MQ water. The seeds were then transferred to germination boxes, where the reservoirs were filled with 25 mL MQ water. The plastic germination boxes, previously sterilized with 70% ethanol, were equipped with two strips of filter paper positioned beneath a square filter paper on a plateau. Each ecotype’s seeds were individually placed in these boxes using a pipette. The germination boxes were then covered with foil to simulate natural dark conditions. The climate chambers were set to match the four test temperatures (10 °C, 17 °C, 24 °C, and 29 °C), and germination was counted at 7 dai, with the use of a magnifying glass. Four independent experiments were conducted to assess germination at each temperature, with each experiment including 36 seeds per temperature treatment.

### 4.3. Bacterial Strains, Seed Inoculation, and Seed Germination Assay at Suboptimal Temperature

In this study, various bacterial strains were used, namely SS101, UW4, FZB, and KT2440, as well as a KT *alg*^−^ mutant and a quadruple mutant KTq (*alg*^−^ *bcs*^−^ *pea*^−^ *peb*^−^), in which all four identifiable putative exopolysaccharide gene clusters responsible for alginate, bacterial cellulose synthesis, putida exopolysaccharide a, and putida exopolysaccharide b were deleted. These strains were obtained from the strain repositories of the Department of Plant and Environmental Sciences at the University of Copenhagen, Denmark. These bacterial strains were sourced from glycerol stock (−80 °C) and cultivated in Lysogeny Broth (LB) medium. Subsequently, they were routinely grown at 28 °C for 12 h to prepare them for various experiments. Liquid bacterial cultures were established by inoculating liquid LB medium with freshly grown bacterial colonies, followed by overnight incubation with agitation at 200 rpm at 28 °C. The bacterial culture was then subjected to centrifugation at 3077 rcf for 10 min, and the supernatant was discarded. The bacterial pellet was resuspended in 10 mM MgCl_2_, and the optical density at 600 nm (OD600) was adjusted to 0.4, equivalent to approximately 3.2 × 10^8^ CFU/mL, for seed inoculation.

Prior to seed inoculation, the seeds were surface-sterilized and stratified at 4 °C for two days. For inoculation, the seeds were dipped into the bacterial suspension and placed in an incubator equipped with an incubator shaker for one hour at 150 rpm, at temperatures ranging from 26 °C to 28 °C. For the control treatment, the seeds were inoculated with 10 mM MgCl_2_ [18]. Subsequently, the seeds were immediately transferred to a germination box, where the reservoir was filled with 25 mL of bacteria or a control treatment, saturating the filter paper. The germination boxes were then covered with foil to simulate natural dark conditions. The climate chambers were set to two suboptimal temperatures (10 °C and 29 °C), and germination was counted at 7 dai with the use of a magnifying glass. To better understand the role of biofilms in germination, the number of germinated seeds of KT2440 and its mutants was counted at 5, 7, and 10 days using a magnifying glass. Four independent experiments were conducted to assess the bacterial impact on germination, each including 36 seeds per treatment.

### 4.4. Statistical Analysis

Statistical analyses were conducted using R version 4.4.0. One-way ANOVA was conducted to determine whether there was a significant difference between each bacterial isolate in *A. thaliana* seed germination. This was followed by a post-hoc Tukey test for multiple comparisons to identify the different bacterial treatments without inflating Type 1 error. The results of the post-hoc Tukey test are then summarized by the compact letter display (depicted as lowercase letters), where groups with no significant difference are groups under the same letter. Polynomial regression models were applied at different time points during the germination series. Data were obtained from three independent experiments, each of which included three biological replicates with 18 seeds per treatment per replicate. Error bars in each figure represent the standard deviation among the independent experiments.

## 5. Conclusions

In conclusion, this study demonstrates the potential of seed biopriming with the biofilm-forming strain *Pseudomonas putida* KT2440 to enhance seed germination at suboptimal low temperatures (10 °C). While biopriming significantly improved the germination of the Cvi-0 ecotype, it did not promote the germination of the widely used Col-0. This study also underscores the important role of biofilm formation in promoting germination under challenging conditions. These findings suggest that PGPR-based biopriming could be a valuable strategy for securing crop yields in the face of climate change and increasing food demands.

## Figures and Tables

**Figure 1 plants-13-02681-f001:**
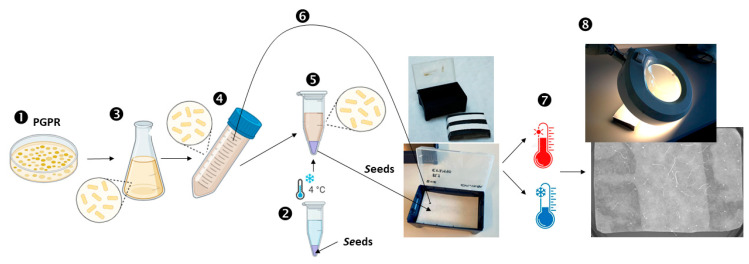
Illustration of the experimental design. (1) Plant Growth-Promoting Rhizobacteria (PGPR) were grown on solid LB plates. (2) Approximately 10 mg of *Arabidopsis thaliana* seeds were sterilized and stratified at 4 °C for two days. (3) Fresh bacterial colonies were inoculated in liquid LB medium and grown overnight. (4) The bacterial culture was centrifuged and the pellet was resuspended in 10 mM MgCl_2_. The optical density at 600 nm was adjusted to 0.4. (5) *A. thaliana* seeds were inoculated with the bacterial suspension; for the control treatment, the seeds were inoculated with 10 mM MgCl_2_. (6) The bacterial suspension and *A. thaliana* seeds were then transferred to a germination box. The box contained an insert surrounded by two filter paper strips to absorb the suspension from the reservoir, with a square filter paper on top where the seeds were placed. (7) The effect of PGPR on *A. thaliana* seed germination was investigated at suboptimal low and high temperatures. (8) Germination was assessed using a magnifying lens.

**Figure 2 plants-13-02681-f002:**
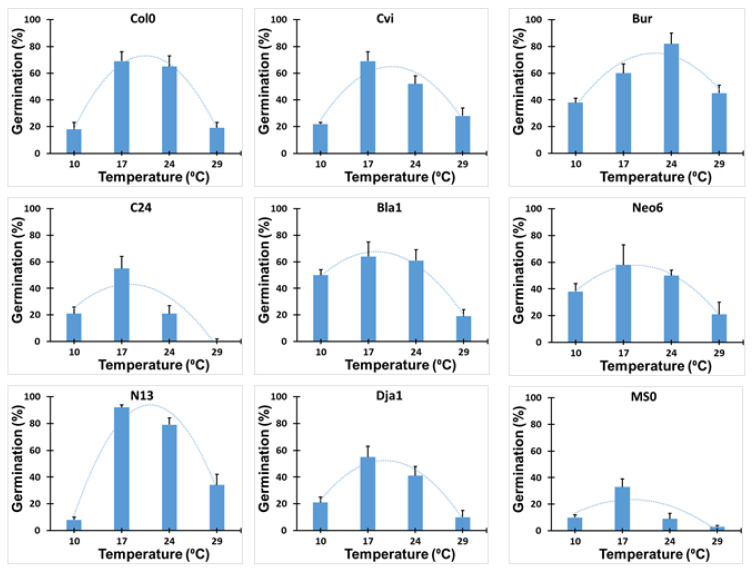
Seed germination percentages at different temperatures for the nine ecotypes. Germination of seeds from nine *A. thaliana* ecotypes observed at different temperatures (10 °C, 17 °C, 24 °C, and 29 °C) after a 7 dai (days after inoculation). Germination was assessed under in-vitro conditions using germination boxes. The highest point on the curve indicates the most favorable temperature for germination specific to each ecotypes, Col0 (Col-0; 19.6 °C), Cvi (Cvi-0; 19.6 °C), Bur (20.6 °C), C24 (17.7 °C), Bla1 (Bla-1; 17.6 °C), Neo6 (Neo-6; 18.2 °C), N13 (20.2 °C), Dja1 (Dja-1; 18.7 °C), and MS0 (17.8 °C). The dotted line represents the polynomial regression, and the fitted polynomial regression equation is shown in Appendix A.

**Figure 3 plants-13-02681-f003:**
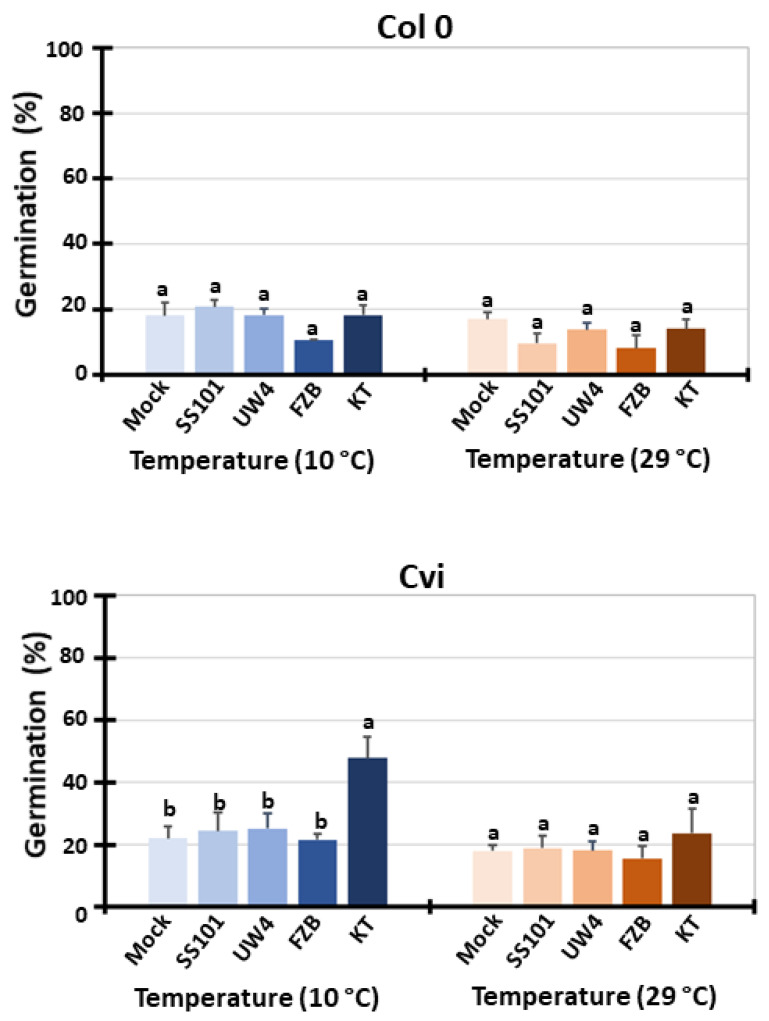
Effect of PGPR on seed germination at suboptimal temperatures (10 °C and 29 °C) in Col-0 and Cvi-0 ecotypes. Seed germination was assessed at suboptimal temperatures (10 °C and 29 °C) for two *A. thaliana* ecotypes in the presence of *Pseudomonas* and *Bacillus* PGPR. Strain KT2440 (KT) showed increased germination in the Cvi (Cvi-0) ecotype under suboptimal low temperatures (10 °C). Significant differences between each bacterial treatment are depicted using different lowercase letters. Error bars in each figure represent the standard deviation of independent experiments.

**Figure 4 plants-13-02681-f004:**
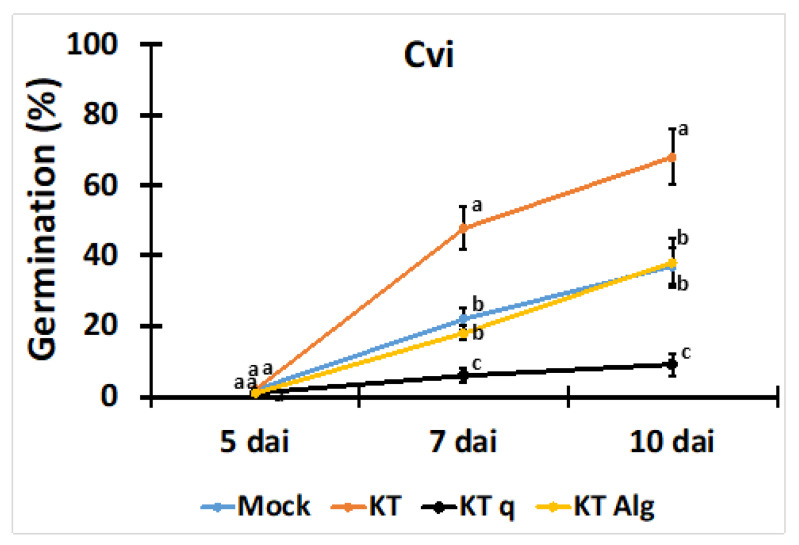
Impact of KT2440 (KT) and its biofilm mutants on seed germination under suboptimal low temperatures (10 °C) in the Cvi-0 ecotype. Seed germination was investigated at a suboptimal low temperature (10 °C) in the presence of biofilm-forming KT2440 and its mutant q/*alg*^−^, with compromised biofilm-forming ability. Significant differences between each bacterial treatment are depicted using lowercase letters. Error bars in each figure represent the standard deviation among the independent experiments.

**Table 1 plants-13-02681-t001:** Geographic origin information for *Arabidopsis thaliana* ecotypes used in this study.

Stock Number	Name	Country	Region Climate	Longitude of Origin	Latitude of Origin	Altitude of Origin	Maximum Temperature	Minimum Temperature	Average Temperature
CS76105	Bur	IRL	Northern Europe Cool & damp	−6.200	54.100	7	15	5.1	9.5
CS76097	Bla-1	ESP	Western Europe Mild & wet	2.800	41.683	117	23.5	11.2	16.7
CS76113	Col-0	USA	Midwest Humid & temperate	−92.300	38.300	229	25.8	0.4	13.4
CS76116	Cvi-0	CPV	Macaronesia Hot & dry	−23.617	15.111	255	26.5	22.7	24.7
CS76106	C24	POR	Southern Europe Dry and temperate	−8.426	40.208	121	20.4	9.1	14.7
CS78246	Dja-1	KGZ	Central Asia Cold	73.633	42.583	1467	10.6	−17.2	−3.3
CS76555	Ms0	RUS	Central Asia Cold	37.350	55.75	137	18.4	−9.7	4
CS76194	N13	RUS	Central Asia Cold	34.150	61.360	219	16	−11.3	1.9
CS76560	Neo-6	TJK	Central Asia Cold	72.467	37.350	3554	15.1	−13.4	1.8

Abbreviations: United States of America—USA; Cape Verdi—CPV; Ireland—IRL; Portugal—POR; Spain—ESP; Tajikistan—TJK; Russia—RUS; Kyrgyzstan—KGZ. The average temperature represents the mean temperature from January to December.

## Data Availability

The original contributions presented in the study are included in the article/Appendix A, further inquiries can be directed to the corresponding author.

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
