# Peer review of "Stimulation of Arabidopsis thaliana Seed Germination at Suboptimal Temperatures through Biopriming with Biofilm-Forming PGPR Pseudomonas putida KT2440"

_plants, 2024, doi:10.3390/plants13192681_

Round 1

Reviewer 1 Report

Comments and Suggestions for Authors

The manuscript entitled “Stimulation of Arabidopsis thaliana Seed Germination at Suboptimal Temperatures through Biopriming with Biofilm-Forming PGPR Pseudomonas putida KT2440” submitted for evaluation promises to be interesting. However, it should be corrected. Editing errors should be corrected, and some passages should be made more detailed.

Authors should ensure the correct order of the chapters in accordance with the instructions. The sections of the research manuscript should be arranged in the following order: Introduction, Results, Discussion, Materials and Methods, and Conclusions. I suggest introducing  conclusions as a separate chapter. References should be prepared correctly according to  instructions. All authors of the publication should be listed. The use of "et al." is incorrect. I advise to read the instructions for authors or use Microsoft Word template to prepare the manuscript.

In my opinion, the title is informative. The abstract is of appropriate length and contains the necessary information. However, I propose to explain the abbreviations used (lines 18, 20, 22). When reading the abstract, readers do not know what this abbreviation means. The abstract must be understandable to the reader without reading the full manuscript. The introduction is well-prepared and concise. The paper presents the general topic and the aim that scientists set themselves when starting their research.

The methodology needs to be supplemented. The reader does not know what criteria the authors used when selecting Arabidopsis ecotypes. It is worth describing their natural habitats (line 89). Please provide origin, region, and climatic conditions, etc. The authors repeatedly refer to ecotypic differences, but nowhere describe them comprehensively. To make point 4.1 of the discussion understandable, it is necessary to describe the Arabidopsis ecotypes. According to me, a mutant that does not produce alginate should be marked with a minus sign, for example, KT alg- (line 107). Also, describe TKQ mutant briefly and correctly (lines 108-109). What does TKQ mean? The reader may not know what polymers this mutant does not synthesize. Please include information in the methodology that seed germination efficiency is expressed as a percentage of seed germination, and the optimal germination temperature was calculated based on the analysis of the peak point of the polynomial curve.

The "Results" chapter needs to be reorganized. The authors should provide a concise and precise description of the experimental results. However, any explanations of the observed phenomena should be included in the discussion. I advise you to avoid repeating methodological information in the description of the results. Verify the information provided on lines 180 and line 190. All repetitions that occur in the manuscript should be eliminated.

Please explain in Section 4.2 why KT2440 as a only stimulated Cvi seed germination. How did it differ from other strains? The authors do not write anything about the possibility of biofilm formation by other strains. What attributes promote plant growth that the bacterial strains used in the research have? Please discuss this. The discussion can be combined with the results if such an arrangement suits the authors. In my opinion, it is advisable to include the final conclusions in a separate chapter.

Detailed comments are included in the pdf file.

Author Response

Responses to the suggestions/comments of Reviewer 1:

2. Questions for General Evaluation

Reviewer’s Evaluation

Response and Revisions

Does the introduction provide sufficient background and include all relevant references?

Yes

Is the research design appropriate?

Yes

Are the methods adequately described?

Must be improved

We have improved the Methods section and added an additional figure illustrating the experimental design.

Are the results clearly presented?

Must be improved

We have also revised the Results section for clarity.

Are the conclusions supported by the results?

Must be improved

Additionally, we have updated the Conclusions section.

Comment 1: The manuscript entitled “Stimulation of Arabidopsis thaliana Seed Germination at Suboptimal Temperatures through Biopriming with Biofilm-Forming PGPR Pseudomonas putida KT2440” submitted for evaluation promises to be interesting. However, it should be corrected. Editing errors should be corrected, and some passages should be made more detailed.

Response 1: We have carefully considered the highly constructive suggestions (highlighted in the file peer-review-38582264.v2.pdf) provided by Reviewer 1 on our manuscript and have incorporated them into the revised version, which is marked in track changes.

Comment 2: Authors should ensure the correct order of the chapters in accordance with the instructions. The sections of the research manuscript should be arranged in the following order: Introduction, Results, Discussion, Materials and Methods, and Conclusions. I suggest introducing  conclusions as a separate chapter. References should be prepared correctly according to  instructions. All authors of the publication should be listed. The use of "et al." is incorrect. I advise to read the instructions for authors or use Microsoft Word template to prepare the manuscript.

Response 2: Thank you for your valuable feedback. We have carefully reviewed the structure of our manuscript and made the necessary revisions to ensure it follows the recommended order: Introduction, Results, Discussion, Materials and Methods, and Conclusions. We have also introduced the Conclusions as a separate chapter as suggested. Regarding the references, we have rechecked them and ensured they are prepared correctly according to the instructions. We have also listed all authors of the publication.

Comment 3: To improve the manuscript's adherence to submission guidelines, we have reviewed the instructions for authors and utilized the Microsoft Word template as advised. In my opinion, the title is informative. The abstract is of appropriate length and contains the necessary information. However, I propose to explain the abbreviations used (lines 18, 20, 22). When reading the abstract, readers do not know what this abbreviation means. The abstract must be understandable to the reader without reading the full manuscript. The introduction is well-prepared and concise. The paper presents the general topic and the aim that scientists set themselves when starting their research.

Response 3: Thank you for your feedback. We have reviewed the instructions for authors and used the Microsoft Word template as advised to improve adherence to submission guidelines. Regarding the abstract, we appreciate your suggestion and have explained the abbreviations used in the abstract. We are pleased to hear that you found the introduction to be well-prepared and concise. It is reassuring to know that the paper effectively presents the general topic and the research aim.

Comment 4: The methodology needs to be supplemented. The reader does not know what criteria the authors used when selecting Arabidopsis ecotypes. It is worth describing their natural habitats (line 89). Please provide origin, region, and climatic conditions, etc. The authors repeatedly refer to ecotypic differences, but nowhere describe them comprehensively. To make point 4.1 of the discussion understandable, it is necessary to describe the Arabidopsis ecotypes. According to me, a mutant that does not produce alginate should be marked with a minus sign, for example, KT alg- (line 107). Also, describe TKQ mutant briefly and correctly (lines 108-109). What does TKQ mean? The reader may not know what polymers this mutant does not synthesize. Please include information in the methodology that seed germination efficiency is expressed as a percentage of seed germination, and the optimal germination temperature was calculated based on the analysis of the peak point of the polynomial curve.

Response 4: We have included the Arabidopsis ecotypes by describing their natural habitats with origin, region, and climatic conditions, etc (P9, line 546-558).

Thank you for pointing this out. We agree that the importance of providing a clear description of the Arabidopsis ecotypes to make our discussion in point 4.1 more understandable. In response to your feedback, we have added a detailed description of the Arabidopsis ecotypes in the revised manuscript (P9, line 546-558). We believe that these additions will provide a comprehensive understanding of the ecotypes and their relevance to our findings, enhancing the clarity of point 4.1 in the discussion section.

We have addressed your concerns regarding the mutant designations and the description of the KTQ mutant in the revised manuscript

We have incorporated the suggested information into the methodology section of our revised manuscript. We believe these revisions address your concerns and improve the overall quality of the manuscript.

Comment 5: The "Results" chapter needs to be reorganized. The authors should provide a concise and precise description of the experimental results. However, any explanations of the observed phenomena should be included in the discussion. I advise you to avoid repeating methodological information in the description of the results. Verify the information provided on lines 180 and line 190. All repetitions that occur in the manuscript should be eliminated.

Response 5: Thank you for your valuable feedback regarding the organization of the "Results" chapter. We have carefully reviewed and revised this section and tried to incorporate your suggestions by focusing on key finding. We removed all part which were related to discussion but we felt we need a short/concise intro to particular experimental approach.

Comment 6: Please explain in Section 4.2 why KT2440 as a only stimulated Cvi seed germination. How did it differ from other strains? The authors do not write anything about the possibility of biofilm formation by other strains. What attributes promote plant growth that the bacterial strains used in the research have? Please discuss this. The discussion can be combined with the results if such an arrangement suits the authors. In my opinion, it is advisable to include the final conclusions in a separate chapter.

Response 6: We have carefully considered your suggestions and made the following revisions to the manuscript: Biofilm Formation by Other Strains: We have added detailed information in Section 4.2 regarding the biofilm-forming abilities of the other bacterial strains used in the study. This includes a discussion of how these abilities compare to those of KT2440, and why KT2440 uniquely stimulated Cvi seed germination. We have also explored the specific attributes of KT2440 that may contribute to its effectiveness, such as its superior biofilm formation and other plant growth-promoting traits. (P10, line 622-625)

Discussion on Plant Growth-Promoting Attributes: We have expanded the discussion to include a comparison of the plant growth-promoting attributes of the bacterial strains used in our research. This includes traits such as biofilm formation, production of phytohormones, nutrient solubilization, and other mechanisms that may contribute to the observed effects on seed germination. This added context helps clarify why KT2440 was particularly effective in stimulating Cvi seed germination compared to the other strains. (P8, line 455-459)

Discussion and Results Structure: We appreciate your suggestion to combine the discussion with the results. However, after careful consideration, we have decided to keep these sections separate to maintain clarity and structure. This separation allows us to present the results in a straightforward manner and then explore into a detailed analysis and interpretation in the discussion.

Separate Conclusions Section: In response to your recommendation, we have included a separate conclusions section. This section summarizes the key findings of our study, emphasizing the unique role of KT2440 in promoting Cvi seed germination and the broader implications of our research. (P11, line 725-732)

Reviewer 2 Report

Comments and Suggestions for Authors

This is an interesting paper and you have some good results. However I have some comments/questions as follows:

Abstract

How does seed biopriming with PGPR have the potential to boost crop yields? How often is crop yield reduced because of poor germination? How can germination be "stimulated". Did you demonstrate enhanced "early growth vigour at low temperature"?

Please rewrite the Abstract and be more accurate.

Introduction.

"transformation of dormant seeds"-dormancy is fleeting or absent in many plant species. Better to say "transformation of quiescent seeds into--". How does germination influence overall plant health? Are you sure about the statement "PGPR are known to boost crop yields significantly"/ What do 1.1 and 1.2 fold actually mean?

"--nutrients---accessible for seed uptake". How does the seed uptake nutrients? The germination process is driven by conversion of stored seed compounds into energy sources; external nutrient resources are not required. Much of the content of this paragraph has nothing to do with germination.

 How does your study "aim to determine temperature dependency of seed germination"? Your wording suggests this is unknown.

Be more clear about your Aims.

Materials and Methods.

2.1. Information on where the ecotypes come from would be useful; using only code names doesn't inform the reader.

 Why did you need to carry out this vigorous seed sterilization process?

 Why were these four temperatures for germination chosen? What temperature is considered optimal for Arabidopsis? What is a germination box?

2.2. Why were the seeds vernalized for two days? How was this done?

2.3. In 2.2 you say the seeds were dipped in bacterial solution and immediately transferred to the germination box. In this section you say the boxes were inoculated by saturating the filter paper with the bacteria. Please clarify.

 You have not clearly distinguished between the method for the genotype/temperature experiment and the method for the PGPR experiment.

 How did you define germination? Was it protrusion of the radicle (physiological germination) or production of a normal seedling (as defined by the International Seed Testing Association)? As you went to 10 days I assume it was the latter? If so, how did you decide what a normal seedling was?

Results.

3.1. Don't repeat Introduction/Methods in the Results section. Your "results" actually start on line 8 of this paragraph.

 What does "germination efficiency" mean?

Why no mention of the significant differences between germination at 17 and 24C for some genotypes (eg Cvi, C24)?

 Five genotypes had higher germination at 10C than 20C, not four.

What happened to the seeds that did not germinate at the suboptimal temperatures? Did they produce abnormal seedlings, did they remain fresh ungerminated, or did they die? You need to provide these data. Or, particularly at 10C, did they have insufficient time to complete the germination process?

3.2.

 Same comments as for 3.1; don't give Introduction/Method in the results section.

 You use the term "germination rate" when your data are correctly "germination percentage". The two are not the same. Please correct.

 You now have included Discussion in the Results!

3.3. The first two paragraphs are Methods, and then after 5 lines of Results you have Discussion again.

4. Discussion.

 The first paragraph is not clear and not required.4.1. "germination increased at temperatures>10C with an optimum between 17 and 24C".This implies that you tested a whole range of increasing temperatures which you did not.Be more carefull about how you word your statements.

 Where did you show that 17 and 24C "promoted higher germination at earlier stages"?

 You say that the duration of the experiment was 7dai, but in the Method you say you went to 10dai.

 Was germination "delayed" at lower/higher temperatures? This statement implies that if you had left the seeds to germinate for longer than 10 days the germination would have increased? There are many germination/temperature experiments which demonstrate that seeds of the same seed lot being set to germinate at low temperature (eg 7C) and optimum temperature (eg 25C) will end up with the same germination percentage, but seeds at the lower temperature will take longer to do so.

4.2. The paragraph on PGPR impacts should concentrate on germination responses only.

 How did you study "establish a standardized germination methodology"? What impact did the seed sterilization have on your rezsults, given that at least for crop seeds this would never occur?

 Why do you think FZB significantly reduced germination of Col0?  Why no responses at 29C?

Your study gave no evidence of the capacity of the PGPR to influence germination; only KT2440 did so. Avoid making sweeping, unsupported statements.

4.3. "KT2440 consistently and significantly improved germination"; again, a sweeping and incorrect statement. KT2440 increased germination   at 10C for one of two genotypes, but for neither at 29C.

 How did KTalg "promote germination"? It didn't differ from the mock control.

 You have produced an interesting result which suggests that biofilm production is the reason why KT2440 was able to increase germination at 10C. This is the majoe finding from this research. I would therefore have expected much more in the Discussion about biofilms. I also wonder why you have made no attempt to explain the different response to KT2440 between the two genotypes.

In the manuscript thefre are too many broad, sweeping statements; please rewrite more accurately.

Comments on the Quality of English Language

The English language quality is on the whole very good. Proof read to correct small errors.

Author Response

Responses to the suggestions/comments of Reviewer 2:

2. Questions for General Evaluation

Reviewer’s Evaluation

Response and Revisions

Does the introduction provide sufficient background and include all relevant references?

Can be improved

We have also revised the introduction and background section.

Is the research design appropriate?

Can be improved

We have added the reviewer's specific recommendations and improved it by adding figure 1.

Are the methods adequately described?

Must be improved

We have improved the Methods section and added an additional figure illustrating the experimental design.

Are the results clearly presented?

Can be improved

We have also revised the Results section for clarity.

Are the conclusions supported by the results?

Can be improved

Additionally, we have updated the Conclusions section.

Comment 1: Abstract: How does seed biopriming with PGPR have the potential to boost crop yields? How often is crop yield reduced because of poor germination? How can germination be "stimulated". Did you demonstrate enhanced "early growth vigor at low temperature"? Please rewrite the Abstract and be more accurate.

Response 1: We appreciate the valuable comments from the reviewer and have reformulated the abstract accordingly.

Given the increasing incidences of drought, particularly during the sowing season in late spring and early summer, biopriming with PGPR has the potential to prevent yield losses in crops. We replaced the term 'boost' because poor germination and seedling establishment are significant contributors to yield losses. While biopriming aims to improve germination-likely by enhancing hydration efficiency and reducing imbibition time but the exact mechanisms have yet to be fully elucidated. (P1, line 12)

Comment 2: Introduction-"transformation of dormant seeds"-dormancy is fleeting or absent in many plant species. Better to say "transformation of quiescent seeds into--". How does germination influence overall plant health? Are you sure about the statement "PGPR are known to boost crop yields significantly"/ What do 1.1 and 1.2 fold actually mean?

Response 2: We have revised the sentence to state, " Plant Growth-Promoting Rhizobacteria (PGPR) are valuable bio-stimulants known to avoid yield losses. PGPR application has demonstrated the potential to increase crop yield, typically ranging between 10% to 20%." This adjustment reflects the reviewer's concern about the impact of PGPR on crop yields. (P1, line 44-45)

Comment 3: Introduction "--nutrients---accessible for seed uptake". How does the seed uptake nutrients? The germination process is driven by conversion of stored seed compounds into energy sources; external nutrient resources are not required. Much of the content of this paragraph has nothing to do with germination.

Response 3: We appreciate the reviewer's observations. It is correct that during the germination process, seeds primarily rely on the conversion of stored internal compounds into energy sources, and external nutrients are not essential at this stage. Regarding the sentence, "Certain bacterial strains have the ability to convert both insoluble inorganic and organic nutrients into soluble forms, thereby making them accessible for seed uptake," we acknowledge that this statement may not directly pertain to the initial stages of seed germination. Instead, this ability of PGPR is more relevant to post-germination plant growth, where the availability of soluble nutrients can enhance seedling development and overall plant health. We have revised the paragraph  (P2, line 77)to focus on the role of PGPR in improving nutrient availability for plant growth beyond the initial germination phase. This adjustment clarifies the context and better aligns the content with the biological processes involved.

Comment 3:  How does your study "aim to determine temperature dependency of seed germination"? Your wording suggests this is unknown. Be more clear about your Aims.

Response 3: Thank you for your valuable feedback. The aim of our study is to investigate how different temperatures affect seed germination across various selected ecotypes of Arabidopsis thaliana. While temperature dependency is documented for the widely used Col0 ecotype, it is less understood for the other selected ecotypes. Surprisingly, the optimum temperature were very similar for all ecotypes tested independent of their natural habitats in distinctly different climatic zones. We evaluate how seed germination varies with temperature changes, focusing on both low and high-temperature extremes. By clarifying these aims, we hope to contribute to a better understanding of temperature dependencies in seed germination and the potential of PGPRs as priming agents in challenging environments

Comment 4:  Materials and Methods- Information on where the ecotypes come from would be useful; using only code names doesn't inform the reader. Why did you need to carry out this vigorous seed sterilization process?  Why were these four temperatures for germination chosen? What temperature is considered optimal for Arabidopsis? What is a germination box?

Response 4: Thank you for your insightful comments. We have made the following adjustments and clarifications in response to your queries:

  • We have updated the detailed information on the geographic origins and climatic conditions of each ecotype used in the study. This additional context will help readers understand the environmental backgrounds of the ecotypes.(P9, line 546-558)
  • The vigorous seed sterilization process was necessary to eliminate surface contaminants and prevent microbial interference during germination experiments. This ensures that any observed effects on germination are due to the experimental treatments rather than external microbial activity.
  • The four temperatures chosen for germination (10°C, 17°C, 24°C, and 29°C) were selected based on known variations in temperature sensitivity among different Arabidopsis ecotypes and to represent a range of suboptimal to optimal conditions. These temperatures cover a broad spectrum to evaluate how germination performance is affected by temperature extremes.
  • The optimal temperature for Arabidopsis germination is generally around 22°C to 24°C. This range supports optimal germination rates (polynomial peak) and seedling growth under controlled conditions.
  • A germination box is a controlled environment chamber used to maintain consistent temperature, and humidity for seed germination experiments.

Additionally, we have included Figure 1 in the revised manuscript to provide a visual representation of the experimental setup and conditions used in the study.

Comment 5: Materials and Methods- 2.2. Why were the seeds vernalized for two days? How was this done?

Response 5: Seeds were stratified for two days to promote uniform germination

Comment 6: 2.3. In 2.2 you say the seeds were dipped in bacterial solution and immediately transferred to the germination box. In this section you say the boxes were inoculated by saturating the filter paper with the bacteria. Please clarify.

Response 6: Seeds were initially dipped in a bacterial suspension to allow for direct exposure to the bacteria. This treatment ensures that the seeds are coated with the PGPR before they are placed in the germination box inoculated by saturating the filter paper with the bacteria. We have included Figure 1 (P3, line139-148) to provide a visual representation of the experimental setup, which further clarifies the process. We hope this explanation resolves any confusion and enhances the clarity of our methodology.

Comment 7: You have not clearly distinguished between the method for the genotype/temperature experiment and the method for the PGPR experiment.

Response 7: We have clarified in (P10, line651-656)

 Comment 8: How did you define germination? Was it protrusion of the radicle (physiological germination) or production of a normal seedling (as defined by the International Seed Testing Association)? As you went to 10 days I assume it was the latter? If so, how did you decide what a normal seedling was?

Response 8: Thank you for your detailed question. In our study, germination was defined based on the protrusion of the radicle (physiological germination). We have included Figure 1 (P3, line139-148)  in the revised manuscript to provide a visual representation of the germination assessment criteria and experimental setup. We hope this clarification addresses your concerns and provides a clearer understanding of our germination assessment methodology. Given the suboptimal temperatures used in our study, we extended the observation period to 10 days to ensure comprehensive germination data. This extended duration allowed us to account for any delays in germination and verify that all seeds had the opportunity to germinate fully.

Comment 9: Results. 3.1. Don't repeat Introduction/Methods in the Results section. Your "results" actually start on line 8 of this paragraph.

What does "germination efficiency" mean?

Why no mention of the significant differences between germination at 17 and 24C for some genotypes (eg Cvi, C24)?

 Five genotypes had higher germination at 10C than 20C, not four.

What happened to the seeds that did not germinate at the suboptimal temperatures? Did they produce abnormal seedlings, did they remain fresh ungerminated, or did they die? You need to provide these data. Or, particularly at 10C, did they have insufficient time to complete the germination process?

Response 9: Thank you for your valuable feedback. We aimed to condense the results sections but found it necessary to provide a brief and concise introduction to the specific experimental approach. We have carefully reviewed and incorporated your suggestions, focusing on the key points raised. Germination efficiency is defined by the change in the number of seeds that germinate compared to the control treatment. For seeds that did not germinate at suboptimal temperatures (including 10°C and 29°C), we observed that they did not develop abnormal seedlings. Instead, these seeds generally showed no signs of germination and were counted as non-germinated in the experiment. We have not tested if the seeds had died. Our main aim was to compare the treatment with the control treatment over the same time frame.

Comment 10: 3.2. Same comments as for 3.1; don't give Introduction/Method in the results section.

 You use the term "germination rate" when your data are correctly "germination percentage". The two are not the same. Please correct.

 You now have included Discussion in the Results!

Response 10: As mentioned for above comment we aimed to condense the results sections but found it necessary to provide a brief and concise introduction to the specific experimental approach of given result section. We have modified the text by removing the term germination rate. We have moved the discussion section from the result section.

Comment 11: 3.3. The first two paragraphs are Methods, and then after 5 lines of Results you have Discussion again.

Response 11: As mentioned for above comment we aimed to condense the results sections but found it necessary to provide a brief and concise introduction to the specific experimental approach of given result section.

Comment 12: 4. Discussion.-The first paragraph is not clear and not required.4.1. "germination increased at temperatures>10C with an optimum between 17 and 24C".This implies that you tested a whole range of increasing temperatures which you did not. Be more carefull about how you word your statements.

Response 12: Deleted as per recommendation (P7, line 367)

Comment 13: Where did you show that 17 and 24C "promoted higher germination at earlier stages"?

You say that the duration of the experiment was 7dai, but in the Method you say you went to 10dai.

Response 13: Yes, it is correct that we counted the germinated seeds at 7 days after incubation (DAI) at different temperatures. To avoid confusion, we have deleted the misleading sentences. We counted the germinated seeds for both the KT2440 wild type and mutant-inoculated seeds at multiple time points, including 5, 7, and 10 DAI.(P10, 654-656)

Comment 14:  Was germination "delayed" at lower/higher temperatures? This statement implies that if you had left the seeds to germinate for longer than 10 days the germination would have increased? There are many germination/temperature experiments which demonstrate that seeds of the same seed lot being set to germinate at low temperature (eg 7C) and optimum temperature (eg 25C) will end up with the same germination percentage, but seeds at the lower temperature will take longer to do so.

Response 14: Thank you for your valuable comment. We acknowledge that the concept of germination delay at suboptimal temperatures is crucial for understanding seed behavior. Our study observed that at suboptimal low temperature, germination was delayed in both the mock and KT mutants compared to the KT2440 wild type. While we assessed germination over a 10-day period, it is possible that extending the observation time might have resulted in an increased number of germinations. For a more comprehensive understanding, future studies could extend the observation period and include a broader range of temperatures to better capture the long-term effects of temperature on seed germination.

Comment 15: 4.2. The paragraph on PGPR impacts should concentrate on germination responses only.

How did you study "establish a standardized germination methodology"? What impact did the seed sterilization have on your rezsults, given that at least for crop seeds this would never occur?

Response 15: We have added the new figure 1 to illustrate experimental design to show the a standardized germination methodology. We have tested our standardized protocol method with different kind of crop seeds such as canola, sugar beet, okra.  (P3, line139-148) The seed sterilization process was use to eliminate surface contaminants and prevent microbial interference during germination experiments. This ensures that any observed effects on germination are due to the experimental treatments rather than external microbial activity.

 Comment 16: Why do you think FZB significantly reduced germination of Col0?  Why no responses at 29C?

Your study gave no evidence of the capacity of the PGPR to influence germination; only KT2440 did so. Avoid making sweeping, unsupported statements.

Responses 16: Thank you for your insightful feedback. We recognize the importance of avoiding broad claims without sufficient evidence and have modified the text accordingly. The reduction in Col0 germination observed with FZB treatment may be due to strain-specific interactions, though the exact mechanism remains unclear. However, this reduction was not statistically significant (One-way ANOVA with Tukey's test) in the revised manuscript. The absence of responses at 29°C suggests this temperature may exceed the optimal range for KT2440 to positively influence germination. We acknowledge that KT2440 was the only strain to significantly affect germination, and we have revised the text to ensure our conclusions are fully supported by the data. (P5, line 220-221)

Comment 17: 4.3. "KT2440 consistently and significantly improved germination"; again, a sweeping and incorrect statement. KT2440 increased germination   at 10C for one of two genotypes, but for neither at 29C.

 How did KTalg "promote germination"? It didn't differ from the mock control.

Response 17: Thank you for your feedback. We have revised the manuscript to address your concerns and provide more precise statements (P7, line 334-335)

Comment 18: You have produced an interesting result which suggests that biofilm production is the reason why KT2440 was able to increase germination at 10C. This is the major finding from this research. I would therefore have expected much more in the Discussion about biofilms. I also wonder why you have made no attempt to explain the different response to KT2440 between the two genotypes.

In the manuscript there are too many broad, sweeping statements; please rewrite more accurately.

Response 18: Thank you for your valuable feedback. We appreciate your feedback of the significance of biofilm production in the observed increase in germination at 10°C. We have revised the Discussion section.(P9, line 532-545)

Round 2

Reviewer 1 Report

Comments and Suggestions for Authors

I am grateful to the authors for implementing the suggested amendments. The revised manuscript is considerably more lucid and intelligible. The results obtained are interest and therefore warrant a presentation of the highest quality. Nevertheless, I am somewhat dissatisfied with the description of the climatic conditions from which the tested Arabidopsis ecotypes originate.

It has come to my attention that not all of my comments and suggestions have been incorporated into the revised manuscript. Nevertheless, I acknowledge and respect the authors' perspective. I am grateful for the clarification provided regarding the rationale behind the decision not to implement the proposed alterations. While perusing the revised manuscript, I discovered several minor errors that must be rectified before printing.

1. Figure 3.  In the initial version of the manuscript, the authors indicated a statistically significant decrease in the germination of Col0 seeds in the presence of the FZB strain at 10oC (significance marked with an asterisk). However, in the revised manuscript (after introducing a letter designation of the significance of the changes), this decrease in the number of germinating seeds is no statistically significant. Please verify that this is the correct designation.

2. Please verify the information presented on line 268. It is my view that the phrase "adapted to a hot and humid climate" should be replaced with "adapted to a hot and dry climate". Please verify this information.

3. The names of the bacterial genera (Pseudomonas and Bacillus) should be written in italics (e.g. lines 323, 325 and 329).

4. The list of references requires correction. This is still incorrect.  Please verify the positions: 1, 2, 4, 5, 7, 8, 10, 13-18, 20, 21, 23-26, 28, 29, 32, 35, 37, 41, 43-47, 49.

To illustrate:

The manuscript contains:  Fiodor, A., et al., Biopriming of seeds with plant growth-promoting bacteria for improved germination and seedling growth. Frontiers in Microbiology, 2023. 14: p. 1142966.

The correct format is as follows: Fiodor, A., Ajijah, N., Dziewit L., Pranaw, K., Biopriming of seeds with plant growth-promoting bacteria for improved germination and seedling growth. Front. Microbiol. (title of the journal in abbreviation and italics) 2023, 14, 1142966.

Furthermore, it is requested that the authors exercise greater care in preparing their response to the review in the future. The revised manuscript comprises 595 lines, and the authors indicating the specific corrections made in lines 622-625 and 725-732. This makes it challenging to ascertain the precise nature of the alterations.

Author Response

Responses to the suggestions/comments of Reviewer 1:

2. Questions for General Evaluation

Reviewer’s Evaluation

Response and Revisions

Does the introduction provide sufficient background and include all relevant references?

Yes

Is the research design appropriate?

Yes

Are the methods adequately described?

Must be improved

We have improved the Methods section and added an additional table and improved the text for greater understanding.

Are the results clearly presented?

Yes

Comment 1: I am grateful to the authors for implementing the suggested amendments. The revised manuscript is considerably more lucid and intelligible. The results obtained are interest and therefore warrant a presentation of the highest quality. Nevertheless, I am somewhat dissatisfied with the description of the climatic conditions from which the tested Arabidopsis ecotypes originate.

Response 1: Thank you for your thoughtful and encouraging feedback. We are pleased that you found the revised manuscript to be more lucid and intelligible. We appreciate your recognition of the quality of the results and your suggestion for a high-quality presentation.

Regarding your concern about the description of the climatic conditions from which the tested Arabidopsis ecotypes originate, we have carefully considered this and have made further improvements. We have added an additional table (Table 1) that provides detailed information about the selected ecotypes, including specific climatic data from their regions of origin (Page 3, line 132-133). Our selection of ecotypes was indeed based on contrasting origins, and we believe that the inclusion of this information enhances the clarity and depth of our study.

Comment 2: It has come to my attention that not all of my comments and suggestions have been incorporated into the revised manuscript. Nevertheless, I acknowledge and respect the authors' perspective. I am grateful for the clarification provided regarding the rationale behind the decision not to implement the proposed alterations. While perusing the revised manuscript, I discovered several minor errors that must be rectified before printing.

Response 2: Thank you for your feedback. We appreciate your understanding and respect for our perspective. We have now revised the manuscript for any minor errors that were overlooked in the revision process to ensure that all errors are corrected before the final version is submitted for printing.

Comment 3: Figure 3.  In the initial version of the manuscript, the authors indicated a statistically significant decrease in the germination of Col0 seeds in the presence of the FZB strain at 10oC (significance marked with an asterisk). However, in the revised manuscript (after introducing a letter designation of the significance of the changes), this decrease in the number of germinating seeds is no statistically significant. Please verify that this is the correct designation.

Response 3: We appreciate your concern and for bringing this matter to our attention. In response to your comment, we reanalyzed the statistical data with the assistance of an associate professor specializing in statistics. We have thoroughly verified that the current version of the manuscript reflects the correct statistical designations. For Figure 3, the revised analysis confirms that the decrease in the germination of Col-0 seeds in the presence of the FZB strain at 10°C is not statistically significant. We have updated the manuscript accordingly and are confident that the current designation is accurate.

Comment 4 : Please verify the information presented on line 268. It is my view that the phrase "adapted to a hot and humid climate" should be replaced with "adapted to a hot and dry climate". Please verify this information.

Response 4: Thank you for pointing out the discrepancy in the sentence. We have revised the phrase on Page 8, line 341 to "adapted to a hot and dry climate," as you suggested. We have verified this information and can confirm that it is consistent with the findings reported by Huang et al. (2014).

Comment 5: The names of the bacterial genera (Pseudomonas and Bacillus) should be written in italics (e.g. lines 323, 325 and 329).

Response 5: We have made the necessary changes to the manuscript as per your recommendation. The names of the bacterial genera (Pseudomonas and Bacillus) have been italicized (e.g., Page 9, lines 397, 399, 403).

Comment 6: 4. The list of references requires correction. This is still incorrect.  Please verify the positions: 1, 2, 4, 5, 7, 8, 10, 13-18, 20, 21, 23-26, 28, 29, 32, 35, 37, 41, 43-47, 49.

To illustrate:

The manuscript contains:  Fiodor, A., et al., Biopriming of seeds with plant growth-promoting bacteria for improved germination and seedling growth. Frontiers in Microbiology, 2023. 14: p. 1142966.

The correct format is as follows: Fiodor, A., Ajijah, N., Dziewit L., Pranaw, K., Biopriming of seeds with plant growth-promoting bacteria for improved germination and seedling growth. Front. Microbiol. (title of the journal in abbreviation and italics) 2023, 14, 1142966.

Response 6:   We have revised the manuscript by correcting the formatting of the references.

Comment 7: Furthermore, it is requested that the authors exercise greater care in preparing their response to the review in the future. The revised manuscript comprises 595 lines, and the authors indicating the specific corrections made in lines 622-625 and 725-732. This makes it challenging to ascertain the precise nature of the alterations.

Response 7:   Thank you for your feedback and for bringing this to our attention. We apologize for the oversight in referencing incorrect line numbers and for any confusion this may have caused. Moving forward, we had make sure that all responses are clear, with accurate references to the correct line numbers in the track change manuscript. We appreciate your understanding.

Reviewer 2 Report

Comments and Suggestions for Authors

Abstract

The Abstract has not been rewritten; only minor adjustments made. The first two sentences are not directly connected to this study and should be deleted.

 I suggest "This study investigated the germination response to temperature of seeds of nine Arabidopsis thaliana ecotypes; 10C and 29C were suboptimal low and high temperatures for all nine ecotypes, even though they originated from regions with diverse climates".

 I suggest "We tested the potential of ---------to stimulate seed germination of two ecotypes  at these suboptimal conditions".

 Germination "efficiency" is a meaningless term- delete "efficiency".

 You did not use other ecoytpes - delete "or from other ecotypes".

 You only found the response at low temperature- the last sentence implies you also found it at high temperature?

Results.

Please delete "efficiency" after "germination". 

"We observed differences in germination of Cvi, C24 and MSO at 24C compared to 17C". What does this mean? I suggest " The germination of Cvi, C24 and MSO was significantly greater at 17C than 24C, while the reseverse occurred for Bur. Germination did not differ between the two temperatures for the other five ecotypes (figure 2).

You have used the words "displayed", "demonstrated" "showed"  - replace with "had".

Figure 3- the caption is repetitious and confusing. Why is the statistics method information included here rather than in 2.4? The same comments apply for Figure 2.4.

 Discussion. The paragraph immediately before 4.3. The sentence you have inserted is a repeat of what you have inserted earlier on the page. And, be careful -you did not show that the strains had all these properties. The way you have worded this without any references implies that you did so.

 You have still not addressed the issue of why KT2440 only provided a germination response in only one of the ecotypes. Why did this occur? What was different between the two ecotypes which may have led to this result?

Your explanation of how the biofilm allowed improved germination is completely unsupported. How do you know it was because it "promoted optimal conditions for germination due to improved seed hydration and stress protection"? Where is your evidence? You did not measure either of these two factors. In a scientific paper you cannot make sweeping statements/claims without supporting evidence.

Materials and Methods.

2.4.1 -last paragraph, second to last sentence; this is incorrect. Explain why you chose only C010 and Cvi for this experiment. In the last sentence, delete "The efficiency of".

2.4.2/3.-to me what you have presented is still confusing. You need to clearly separate the methodology for the germination x temperature experiment (results in Fig 2) and the bacteria x suboptimal temperature experiment (results in Fig 3 and 4). For the first experiment, presumably the reservoir  in the germination box contained water only? For the bacterial experiments, how many CFUs/ml of the bacteria were in the reservoir solution?

 Conclusions.

 First sentence- change to "--- germination at a suboptimal low temperature (10C)."

 Second sentence - you only used two ecotypes, so correct this sentence

Comments on the Quality of English Language

The manuscript requires careful editing to correct minor errors of sentence construction, tense and spelling.

Author Response

Responses to the suggestions/comments of Reviewer 2:

2. Questions for General Evaluation

Reviewer’s Evaluation

Response and Revisions

Does the introduction provide sufficient background and include all relevant references?

Yes

Is the research design appropriate?

Yes

Are the methods adequately described?

Can be improved

We have improved the Methods section and added an additional table and also improved the text for greater understanding.

Are the results clearly presented?

Can be improved

We have improved the result section for clearity.

Comment 1:  Abstract:

The Abstract has not been rewritten; only minor adjustments made. The first two sentences are not directly connected to this study and should be deleted.

 I suggest "This study investigated the germination response to temperature of seeds of nine Arabidopsis thaliana ecotypes; 10C and 29C were suboptimal low and high temperatures for all nine ecotypes, even though they originated from regions with diverse climates".

 I suggest "We tested the potential of ---------to stimulate seed germination of two ecotypes  at these suboptimal conditions".

 Germination "efficiency" is a meaningless term- delete "efficiency".

 You did not use other ecoytpes - delete "or from other ecotypes".

 You only found the response at low temperature- the last sentence implies you also found it at high temperature?

Response 2: We have revised the abstract as suggested. (Page 1, line 11-16, 20, 22, 26

Comment 2:   Results. Please delete "efficiency" after "germination". 

Response 2:  We have made the necessary changes to the manuscript as per your recommendation. The term 'efficiency' after germination has been removed.

Comment 3:   We observed differences in germination of Cvi, C24 and MSO at 24C compared to 17C". What does this mean? I suggest " The germination of Cvi, C24 and MSO was significantly greater at 17C than 24C, while the reseverse occurred for Bur. Germination did not differ between the two temperatures for the other five ecotypes (figure 2).

Response 3: Thank you for your suggestion. We have revised the sentence as per your recommendation. The updated text now reads: "The germination of Cvi, C24, and MSO was significantly greater at 17°C than at 24°C, while the reverse occurred for Bur. Germination did not differ between the two temperatures for the other five ecotypes (Figure 2). (Page 4, 163-165)

Comment 4:   You have used the words "displayed", "demonstrated" "showed"  - replace with "had".

Response 4: Thank you for your feedback. We have made the changes to the manuscript as recommended. The words "displayed," "demonstrated," and "showed" have been replaced with "had" where appropriate.

Comment 5:   Figure 3- the caption is repetitious and confusing. Why is the statistics method information included here rather than in 2.4? The same comments apply for Figure 2.4.

Response 5: We have shortened the text in the figure captions to avoid the repetition. This statistical methods have been removed from the captions and are described in Section 4.4, as per your recommendation.

Comment 6:   The paragraph immediately before 4.3. The sentence you have inserted is a repeat of what you have inserted earlier on the page. And, be careful -you did not show that the strains had all these properties. The way you have worded this without any references implies that you did so.

Response 6: Thank you for your feedback. We have revised the paragraph by removing the repeated sentence and clarifying the statement about the strains' properties to ensure it does not imply unsubstantiated claims. Appropriate references have been included where necessary (Page 9, line 407-411).

Comment 7: You have still not addressed the issue of why KT2440 only provided a germination response in only one of the ecotypes. Why did this occur? What was different between the two ecotypes which may have led to this result?

Response 7: Thank you for your valuable feedback. We acknowledge that the manuscript does not address the reason why strain KT2440 stimulated germination of only one of the two tested ecotypes. The difference in germination response could have various reasons and potentially be related to intrinsic differences between the two ecotypes related to their distinctly different habitats characterized by specific pedoclimatic conditions. The differences could potentially be related to variations in seed coat permeability, metabolic profiles, or hormonal balances, which were not measured in this study. Ecotypes often exhibit different physiological and biochemical traits that are expected to also influence their biotic interactions such as with beneficial bacteria. For example, certain ecotypes may have seed coats with differing permeability, affecting how substances like water or bacterial signals interact with the seed. Another possible factor could be differences in the present seed endophytes, which might interact with KT2440 in diverse ways. Due the many possible reasons for the differential response the elucidation of the underlying mechanisms would be a study of its own. However, the finding of ecotype specific differences shows that genotypic differences within a particular species may determine a differential responsiveness to beneficial microbes. Thus, this conclusions is relevant for the development of robust biostimulants for crop plants. One major problem of biostimulants is the lack of robustness across cultivars of a particular crop species. Thus, the observed differences in responsiveness could be used as basis to identify the genetic determinants which then can be established as novel breeding target to improve responsiveness across cultivar. The corresponding discussion had been added at the end of the manuscript. (Page 10, line 485-495)

Comment 8: Your explanation of how the biofilm allowed improved germination is completely unsupported. How do you know it was because it "promoted optimal conditions for germination due to improved seed hydration and stress protection"? Where is your evidence? You did not measure either of these two factors. In a scientific paper you cannot make claims without supporting evidence.

Response 8: Regarding the claim that the biofilm "promoted optimal conditions for germination due to improved seed hydration and stress protection," we acknowledge the reviewer's point that this statement lacks direct supporting evidence. We agree that this statement was speculative and needs to be revised. To address this, we have revised the manuscript to removed unsupported claims or backed up claims with related references. We emphasize that while we have functionally shown with a genetic approach with two different types of mutants that biofilm formation is relevant for the stimulation of seed germination we can only speculate about the underlying mechanism how the observed effects are mediated. There was a clear correlation between the number of mutated biofilm relevant bacterial genes and the reduction in the ability to stimulate seed germination. We also discuss the need for future experiments to address the underlying mechanisms by directly measuring factors such as seed hydration, stress responses, and the biochemical interactions between KT2440 and the seeds, in order to better understand the role how biofilms mediated the process of the stimulation of seed germination.

Comment 9:   Materials and Methods. 2.4.1 -last paragraph, second to last sentence; this is incorrect. Explain why you chose only C010 and Cvi for this experiment. In the last sentence, delete "The efficiency of".

Response 9: We have revised the text to accurately explain our rationale for selecting only Col-0 and Cvi for further investigation. These ecotypes were chosen due to Col0's widespread use as a reference species in research and Cvi's unique adaptation to hot, dry climates, making them ideal candidates for examining the effects of PGPR under temperature stress conditions at both 10°C and 29°C. Additionally, we found that both Col0 and Cvi had the same favorable temperature for germination, with the peak at 19.6°C, which further justified their selection. We have also deleted the phrase "The efficiency of" as you suggested. (Page 15, line 224-227)

Comment 10:   2.4.2/3.- You need to clearly separate the methodology for the germination x temperature experiment (results in Fig 2) and the bacteria x suboptimal temperature experiment (results in Fig 3 and 4). For the first experiment, presumably the reservoir in the germination box contained water only? For the bacterial experiments, how many CFUs/ml of the bacteria were in the reservoir solution?

Response 10: Thank you for your valuable feedback. We have revised the text to clearly separate the methodology for the germination × temperature experiment (results in Figure 2) from the bacteria × suboptimal temperature experiment (results in Figures 3 and 4). We have added all relevant information to the Materials and Methods section. For the germination × temperature experiment, we confirm that the reservoir in the germination box contained water only. For the bacterial experiments, we used a concentration of approximately 3.2×108 CFUs/ml in the reservoir solution (P11, Line 564).

Comment 11:    Conclusions. First sentence- change to "--- germination at a suboptimal low temperature (10C)." Second sentence - you only used two ecotypes, so correct this sentence

Response 11: Thank you for your feedback. We have revised the first sentence of the Conclusions section as per your recommendation, changing it to "germination at a suboptimal low temperature (10°C)." Additionally, we have corrected the second sentence to accurately reflect that we used only two ecotypes in our study. The revised sentence now reads: "While the biopriming significantly improved the germination of the Cvi ecotype, it did not promote the germination of the widely used Col0” (Page 12, Line 748-750)

Additionally, we have thoroughly reviewed the manuscript to correct minor errors related to sentence construction, tense, and spelling. We appreciate your patience and understanding, and we are committed to maintaining greater precision and clarity in our responses moving forward.